# Phytoplankton Retention Mechanisms in Estuaries: A Case Study of the Elbe Estuary

Laurin Steidle[1] and Ross Vennell[2]

[1]Institute of Marine Ecosystem and Fishery Science, Universität Hamburg, Olbersweg 24, 22767 Hamburg, Germany
[2]Cawthron Institute, 98 Halifax Street East, Nelson 7010, New Zealand

**Correspondence:** Laurin Steidle (laurin.steidle@uni-hamburg.de)

**Abstract.** Due to their role as primary producers, phytoplankton are essential to the productivity of estuarine ecosystems. However, it is important to understand how these nearly passive organisms are able to persist within estuaries, when river inflow results in a net outflow to the ocean. Estuaries are also representing challenging habitats due to a strong salinity gradient. So far, little is known about how phytoplankton are able to be retained within estuaries. We present a new individual-based Lagrangian model of the Elbe estuary which examines possible retention mechanisms for phytoplankton. Specifically, we investigated how reproduction, sinking and rising, as well as diel vertical migration may allow for populations to persist within the estuary. We find that vertical migration especially rising favors the retention, fast sinking does not. We further provide first estimates on outwashing losses. Our simulations illustrate that riverbanks and tidal flats are essential for the long-term survival of phytoplankton populations, providing refuges from strong downstream currents. These results contribute to the understanding needed to advance ecosystem-based management of estuaries.

## 1 Introduction

Estuaries are highly productive ecosystems. Their relative small area disproportionally contributes to the global carbon cycle, along with their role as a source of nutrients and hatching grounds for marine ecosystems (Cloern et al., 2014; Arevalo et al., 2023). While they are heavily influenced by anthropogenic stressors such as diking, dredging, and fishing, they are of tremendous importance for anthropogenic usage (Jennerjahn and Mitchell, 2013; Brown et al., 2022; Wilson, 2002). Estuaries present challenging dynamics to their smallest residents, due to strong salinity gradient and a net transport to the ocean. Here, we explore how phytoplankton, drifting small primary producers that form the basis of estuarine food webs, can persist within such dynamic environments.

Like most ecosystems - estuarine ecosystem dynamics are strongly controlled by primary producers, in particular phytoplankton (Chen et al., 2023). Apart from biofilm-forming phytoplankton which are attached to their substrate (Cheah and Chan, 2022), the vast majority of phytoplankton organisms drifts passively in currents, though they may be able to influence their vertical movement. With the estuary having a net outwards flow, we would expect phytoplankton to be moving downstream over time and to be washed out from limnic waters, via brackish into marine waters. Hence, the question arises how phytoplankton, as the drifting base of estuarine food webs, are able to maintain their population size without declining due

to the net transport into the open ocean. If we assume that the population is not exclusively maintained by a self maintaining source population upstream, that is washed into the estuary, then there must be some sort of retention mechanism that enables a phytoplankton population to persist within the estuary.

So far, different theories exist on how estuarine phytoplankton populations are able to maintain their position. Previous observational studies suggested several possible mechanisms that could enable retention of phytoplankton populations within estuarine systems - vertical migration in the form of sinking, rising or diel migration, stickiness.

Diel vertical migration is a process where organisms move up and down in the water column in response to the sun. This movement may favors retention by allowing plankton to reduce the time in the faster downstream currents at the water surface. A study by Anderson and Stolzenbach (1985) showed that diel migrating dinoflagellates were able to out compete other non-motile phytoplankton in an embayment environment and even compensate for outwashing losses through reproduction increasing their abundance. However, this also implies that the growing part of the population is somehow retaining their position. If the regrowing population is also continuously drifting downstream they will not able to sustain their population in that area and ultimately die out due to unfavorable salinity conditions in marine waters (Admiraal, 1976; von Alvensleben et al., 2016; Jiang et al., 2020). The presence of diel migration has mostly been demonstrated for motile phytoplankton such as dinoflagellates (Hall et al., 2015; Crawford and Purdie, 1991; Hall and Paerl, 2011) and zooplankton species (Kimmerer et al., 2002). While the motivation for diel migration for autotrophic, mixotrophic, and heterotrophic differs, the consequence remains the same, an upward movement during the day and a downward movement during the night.

Estuaries are complex and strongly dynamic systems such that it is still difficult to predict ecosystem dynamics or the effects of anthropogenic impacts due to their complex bathymetry (MacWilliams et al., 2016; Fringer et al., 2019). Nevertheless, there are sophisticated estuarine models that are able to reproduce the complex dynamics of estuaries reasonably well. This includes currents and water levels on the physical side, but also chlorophyll concentrations and other biologically driven properties (Pein et al., 2021; Schöl et al., 2014). However, these are Eulerian models. This means that they are based on a fixed grid and calculate the concentration of a tracer, such as phytoplankton, at each grid cell. This makes it difficult to study concepts such as retention times, as they lack temporal consistency, meaning that the life history and trajectory of a phytoplankton cell cannot be tracked. Previous modeling studies have attempted to overcome this problem using a Lagrangian approach. A Lagrangian model does not try to track e.g. concentrations at fixed positions, but rather follows the motion of individual particles that can be used to represent e.g. water parcels or organisms. Their ability to resolve the interactions of individual phytoplankton cells or aggregates with the bathymetry, e.g. through settling or stranding, while maintaining temporal consistency, is essential for investigating retention mechanisms.

In Simons et al. (2006); Kimmerer et al. (2014) they used a Lagrangian model to study zooplankton retention. Simons et al. (2006) examined the dispersal and flushing time of mussel larvae in the St. Lawrence Estuary while() (Kimmerer et al., 2014) examined zooplankton movement in the San Francisco Estuary. They were able to show that sinking and diel vertical migration slows the outwashing process and might be a beneficial retention strategy. However, they did so by ignoring key processes like reproduction, mortality, and stranding or sedimentation processes. Moreover, both studies were based on low

resolution structured grid models, which we suspect to under-represent the complex bathymetry of estuarine systems (Ye et al., 2018).

Diatoms or benthic microalgae in particular have been observed to be strongly negatively buoyant, hence sinking to the riverbed and remaining there for a long time (Passow, 1991; Thomas Anderson, 1998). Studies also found sticky compounds in phytoplankton aggregates that are suspected to allow them stick to suspended particles, enabling them to sink to the riverbed or sticking to their surroundings aiding retention (Kiørboe and Hansen, 1993; van der Lee, 2000).

In summary, different retention mechanisms have been observed or examined in modeling studies. However, they did so either in isolation in the case of observational studies or with major simplifications in the modeling studies. There is currently a lack of theoretical studies that allow for a more comprehensive overview into the interplay of vertical migration and reproduction in combination with settling and stranding as retention mechanisms.

Here, we explore possible retention mechanisms of phytoplankton using the Elbe estuary as a case study. It is located in the north of Germany and flows into the North Sea. Like most alluvial estuaries, it is relatively shallow, with most of it averaging only a few meters in average depth. Similar to other European estuaries it experienced a strong anthropogenic pressure over the last centuries. Most notably diking to restrain it to a narrow channel and dredging to improve access to the Hamburg harbor. Unlike other major European ports, the port of Hamburg is located far inwards roughly 100 km behind the coastline. To create port access the main channel is dredged and experiences a sudden jump in bathymetry from approximately 5 m at border of the city to up to 20 m in the port and downstream (see fig. 1). This bathymetric jump is suspected to be the cause of a collapse of the phytoplankton, resulting in an increase in oxygen depletion and high ammonium remineralization downstream of the bathymetric jump (Schroeder, 1997; Holzwarth and Wirtz, 2018; Sanders et al., 2018). Ongoing dredging is being carried out to maintain the depth of the navigational channel causing high turbidity (Kappenberg and Grabemann, 2001). While important aspects of the along-channel biochemical dynamics have been studied, little is known about their vertical and shore-to-shore dynamics (Goosen et al., 1999; Dähnke et al., 2008; Sanders et al., 2018).

For this purpose, we further developed the individual-based Lagrangian model OceanTracker (Vennell et al., 2021) and applied it to the Elbe estuary using the hydrodynamics calculated by a recent model SCHISM (Pein et al., 2021). While the Lagrangian model simulated the movement of the inanimate organism, we included key phytoplankton features such as reproduction and mortality, sinking and rising, as well as diel vertical migration. Using this model, we investigate under which conditions phytoplankton retention can be reproduced.

## 2 Methods

### 2.1 Model description

In our study we use a Lagrangian approach with the particle tracking model OceanTracker (Vennell et al., 2021). While off-line particle tracking on unstructured grids has been relatively computationally expensive until recently (Vennell et al., 2021), it offers several advantages. Firstly, it allows us to reuse computationally expensive hydrodynamic models to model tracer-like objects. This is overall much faster than recalculating the advection-diffusion-equation in an Eularian model. Secondly, because

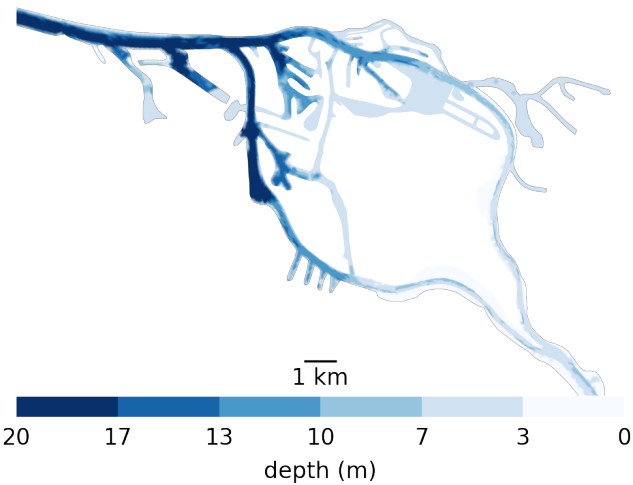

**Figure 1.** Bathymetry of the Elbe model around Hamburg. Note the bathymetric jumps from 5 m on the right, upstream side to a short 10 m step in the upper port area to the 20 m in the lower port area all the way to the North Sea. Also note that there exists no channel that does not pass through the 10 km of exclusively 20 m deep channel.

we are simulating individually particles we are able to observe their tracks. In our model, we use these particles to represent phytoplankton cells. Alternatively, these particles could also be interpreted as aggregates colonized by phytoplankton. The temporal consistency of a Lagrangian model, the fact that we know the history of each particle, makes the interpretation of our results more intuitive and allows us to include individual-based properties and processes that cannot be represented in Eulerian models like e.g. retention times.

We use the hydrodynamic data generated by the latest SCHISM model of the Elbe estuary (Pein et al., 2021) from the weir at Geesthacht to the North Sea, including several side channels and the port area (see figure 2). SCHISM solves the Reynolds-averaged Navier-Stokes equations on unstructured meshes assuming hydrostatic conditions with a time step of 60 s. The unstructured mesh is three-dimensional and consists of 32k horizontal nodes using terrain-following coordinates based on the LSC2 technique (Zhang et al., 2016) for the vertical grid, allowing a maximum number of 20 levels. Regions with depths less than 2 m are resolved by only one vertical level. Bathymetric data was provided by the German Federal Maritime and Hydrographic Agency (Bundesamt fuer Seeschifffahrt und Hydrographie, BSH) and the German Waterways Agency (Wasserstraßen- und Schiffahrtsamt, WSA) with a horizontal resolution of 50 m in the German Bight, 10 m in the Elbe estuary and 5 m in the Hamburg port Stanev et al. (2019). The boundary conditions on the seaward side include sea surface elevation, horizontal currents, salinity and temperature Stanev et al. (2019) and discharge and temperature from the Elbe river on the land-ward side. Atmospheric forcing includes wind, air temperature, precipitation, shortwave and longwave radiation Stanev et al. (2019). Model validation is based on tide gauge stations and long-term stationary measurements of salinity, water temperature, and horizontal currents. Biochemical variables, including chlorophyll, are based on long-term measurements at the Seemannshöft and Grauerort stations Pein et al. (2021). The model provides us with a node-based mesh containing a range

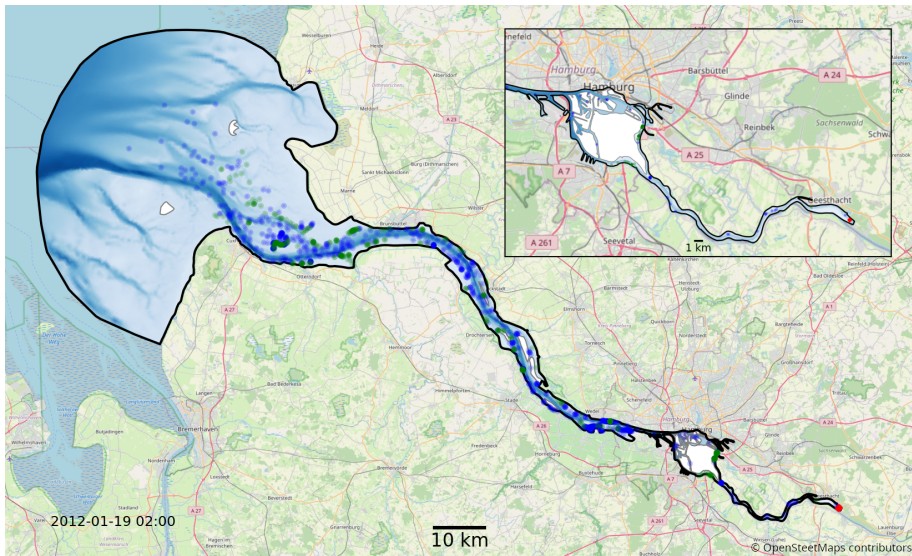

**Figure 2.** Map of the full model domain, with Geesthacht being the upstream boarder on the right and the North-sea being the downstream border on the left. The black outline marks the edge of the model domain. Blue and green dots show an example snapshot of a fraction of the phytoplankton in the model. The location of the initial release is shown in red. Blue represents floating, green phytoplankton stranded by the receding tide. The red area is the initial release location. The background map has been provided by © OpenStreetMap contributors 2023. Distributed under the Open Data Commons Open Database License (ODbL) v1.0.

of information such as water velocity, salinity, water level and dispersion. The year represented in that dataset is 2012 with a temporal resolution of 1 hour and a dynamically varying spacial resolution with distance between nodes ranging from 5 to 1400 m with a median distance of approximately 75 m.

We add a set of biological features on top of the otherwise inanimate organism. These features include reproduction and
mortality, vertical movement in form of sinking, rising or diel vertical migration, stranding, and settling on the riverbed.

Reproduction is represented as a fission process, where each phytoplankton cell has a probability to split effectively producing a copy. This is a novel feature applied in OceanTracker that has not been included in any previous Lagrangian model of this type. OceanTracker's recent advances in computational efficiency (Vennell et al., 2021) and buffer handling make it possible for the first time to simulate a large number of particles over a long period of time on unstructured grids. We perform multiple
simulations for a range of reproduction rates, implemented as a fission probability evaluated every minute, that are constant over the lifetime of the cell. While a fixed reproduction rate is a simplification that does not allow for more realistic simulation of the population dynamics of a particular species, it does allow us to investigate the general mechanisms that enable plankton retention.

Mortality is induced by one of three processes: high salinity, when they dry out while stranded, or due to long term light
limitation. When particles cells are exposed to high salinity water above 20PSU, a mortality probability of $0.5\%$ per minute is applied removing dead phytoplankton cells from the simulation (see salinity map in fig. C1 ). This threshold is chosen based

on a range of the salinity tolerances of estuarine phytoplankton species presented in (von Alvensleben et al., 2016). This is only an approximation and salinity tolerances of many estuarine phytoplankton species deviate from this. However, the main motivation for this choice is that most of the phytoplankton cells that die through this process have passed the isohaline for

more than 12 hours, one tidal cycle, and are assumed not to return again through this isohaline. Anything outside the 20 PSU isohaline is not considered part of the estuary for the purposes of this study. Therefore, we are not tailoring our salinity tolerance to a specific species, but rather testing whether they can retain themselves within this isohaline. We consider phytoplankton cells that are stranded out of the water by the receding tide, and lie dry for more than 7 consecutive days to be dead and remove them. Note that these dry cells are not typically devoid of water, but are considered "dry" if the majority of their area

has a water level below 0.1 m. Additionally, in nature these areas typically contain small sub-resolution structures such as tidal ripples or small puddles, and vegetation that allows these areas to remain wet for periods longer than one tidal cycle. There are currently no studies investigating the time range for survival of stranded phytoplankton on tidal-flats or marshes in estuaries. Therefore, we performed a sensitivity analysis to determine the effect of this parameter on the retention success of the phytoplankton population (see section A). Phytoplankton cells will also die if they are light-limited for 14 days. This value

is based on measurements presented in (Walter et al., 2017) which imply that the majority of phytoplankton is dead after 14 days of light limitation. A sensitivity analysis for this parameter is presented in sec. B. They are considered light-limited below a depth of 1m estimated with the Beer–Lambert law based on SPM data presented in (Stanev et al., 2019). The initial batch of phytoplankton cells starts their life with a full light budget of 14 days, and each minute below 1m reduces this budget by one minute, while the opposite applies if they are above 1m. When a cell splits both inherit the same remaining light budget.

We investigate the effect of different patterns of vertical motion. The first is monodirectional upward or downward vertical motion, representing either positively or negatively buoyant phytoplankton. This buoyancy can be interpreted either as an active choice of buoyancy by the organism through adaptation, or as governed by the suspended matter aggregate on which they live. For monodirectional vertical motion, we assign each phytoplankton cell a vertical velocity, which remains constant throughout its lifetime. The second mode of vertical motion is diel vertical migration. Here phytoplankton cells change their direction of

motion based on the current phase of the sun, creating a motion pattern where they rise during the day and sink during the night. This behavior is often assumed to be performed to maximize light capture while avoiding predation - or, as we suspect, to increase retention.

We include a settling and resuspension model to represent tidal stranding and phytoplankton cells settling on the bed of the estuary. Stranding phytoplankton and microphytobenthos have been shown on several occasions to be a major driver of

estuarine primary production (Carlson et al., 1984; De Jonge and Van Beuselom, 1992; Kromkamp et al., 1995; Savelli et al., 2019). Phytoplankton cells become stranded when the current grid cell becomes dry and stay in place until they are resuspended or dry-out. They are not allowed to move from wet cells to dry cells, by the random walk diffusion applied to all phytoplankton cells. A grid cell is considered "dry" based on the flag given in the SCHISM hydrodynamic model output. Once this grid cell is flooded again, all the stranded phytoplankton cells are resuspended and able to move again. Phytoplankton cells settle on

the bed once they attempt to move below the bottom model boundary and are resuspended based on a critical sheer velocity of

0.009 ms$^{-1}$. The velocity profile in the bottom layer, or log layer, is calculated by

$$U(z) = \frac{u_*}{\kappa} ln \frac{z}{z_0}, \tag{1}$$

where U is the friction velocity representing the drag at height z above the seabed, $\kappa$ is the van Karman constant, $z_0$ is a length scale reflecting the bottom roughness, and $u^*$ is the critical friction velocity. If the friction velocity is above the critical friction velocity the phytoplankton cell is resuspended. Phytoplankton cells that are stranded or settled on the bed are allowed to reproduce. Phytoplankton cells are not only advected but also diffused based on eddy diffusivity which is crucial to represent tidal-pumping processes. Diffusion was modeled using a random walk using a random number generator with a normal distribution. Horizontally the standard distribution of the random walk is set to 0.1 ms$^{-1}$. The vertical displacement of a phytoplankton cell $\partial z\ i$ is calculated by

$$\partial z_i = K_v^{'}(z_i(n))\partial t + N(0, 2K_v(z_i)) \tag{2}$$

based on Yamazaki et al. (2014) where $z_i$ is the vertical position of the phytoplankton cell, $K_v^{'}$ is the vertical eddy diffusivity gradient, $K_v$ is the vertical eddy diffusivity and $N$ is the normal distribution. The term based $K_v^{'}$ is needed to avoid phytoplankton accumulation on the top and bottom of the water column from the hydrodynamic model output.

For each phytoplankton cell we log their distance traveled, age, water depth, and status (whether they are drifting or settled on the river bank or bottom). This allows us for example to compare successfully retained phytoplankton cell (older than three months) unsuccessfully retained phytoplankton cell (dead after less than three months). These observables are recorded every 12 hours starting at midnight.

Model simulations and visualizations were performed in Python making heavy use of Numba, a LLVM-based Python JIT compiler (Lam et al., 2015) to significantly speed up the simulations (Vennell et al., 2021). Trajectories were calculated using a second order Runge-Kutta scheme with a fixed time step of 60 s. Flow velocities, like any other hydrodynamic data, were interpolated linearly in time, linearly in space on the vertical axis and on the horizontal axis using barycentric coordinates, with the exception of water velocity in the bottom cell, where logarithmic vertical interpolation is used to represent drag forces more accurately.

## 2.2 Experimental configurations

We perform two sets of experiments to test the influence of different vertical movements on the retention success of phytoplankton in the Elbe estuary.

In the first experiment, we examine a range of different monodirectional upward or downward particle velocities from $-10$ to $+10$ mm s$^{-1}$ in 2 mm s$^{-1}$ steps representing sinking or rising phytoplankton organisms (Fennessy and Dyer, 1996). Each vertical velocity is examined for a range of different reproduction rates, expressed as population doubling times ranging from 40 to 404 days with a logarithmic scaling. In the following, we will use reproduction rate to refer to the prescribed population growth rate under idealized conditions and use growth rate whenever we describe population growth in nature. The prescribed population growth rate can be interpreted as potential average net-doubling-times in the presence of predation and mortality,

nutrient availability while testing the effect of outwashing. In the second set of model experiments, we study the influence of possible diel vertical migration patterns for the same vertical velocities and reproduction rates. Hence, a total of 187 different scenarios are tested.

In both sets of experiments, we release 10,000 individuals representing a subset of the the studied phytoplankton population at the beginning of the year. This results in over 1 billion individual particles simulated for each case with approximately 1 million particles active simultaneously counted over all cases for 500,000 time steps. This corresponds approximately to a one to one ratio of simulated phytoplankton cells to mesh nodes in the hydrodynamic model at each time step. The initial population is homogeneously distributed in a volume covering the full water column at the weir in Geesthacht (see fig. 2) and examine how the population distributes itself over the estuary and whether it is able to maintain its population size over time. Conceptually, we consider a population to be successfully retained if it is able to sustain itself long term or even shows growth. Practically, this is evaluated by comparing the population size at the end of the year to the size after release. The choice of one year is considered reasonable because it covers the full seasonal cycle and is also much longer than the average exit or flushing time of the estuary (see fig. 6). The first three months of the simulations are considered as initial model spin-up time during which the initial population is dispersed downstream throughout the estuary. Population size changes are measured at the end of the year relative to the population size after this initial spin-up time.

Computations were performed on the supercomputer Mistral at the German Climate Computing Center (DKRZ) in Hamburg, Germany. The simulations were performed on a compute node with two Intel Xeon E5-2680 v3 12-core processor (Haswell) and 128 GB of RAM with a total run time of approximately 4.5 hours.

## 3    Results

### 3.1    Retentions success in different scenarios

The results of the retention experiments are visualized as heatmap in fig. 3. Fig. 3a shows the results for the monodirectional vertical migration scenarios i.e. constant sinking or rising. Fig. 3b shows the results for the diel vertical migration scenarios. Each pixel in the heatmap represents a simulation with a specific combination of vertical velocity and reproduction rate expressed as a population doubling time. The coloring indicates the relative population change after one year. White pixels and the boundary between green and brown pixels represent net-zero growth rate simulations. In this case, the losses are equal to the growth. Therefore, we can use the reproduction rate as an estimate for the total relative losses due to downstream transport, drying out while being stranded, and light starvation.

Our simulations show that the population is able to successfully retains itself under certain conditions. Passively drifting phytoplankton is able to sustain themselves in the estuary if they have a reproduction rate that doubles their population size within approximately 3 months (see fig. 3). Note that the growth rates realized in nature may vary from this value due to e.g. nutrient or temperature limitations. The reproduction thresholds should be interpreted as an upper bound rather than an accurate estimate of the growth rate.

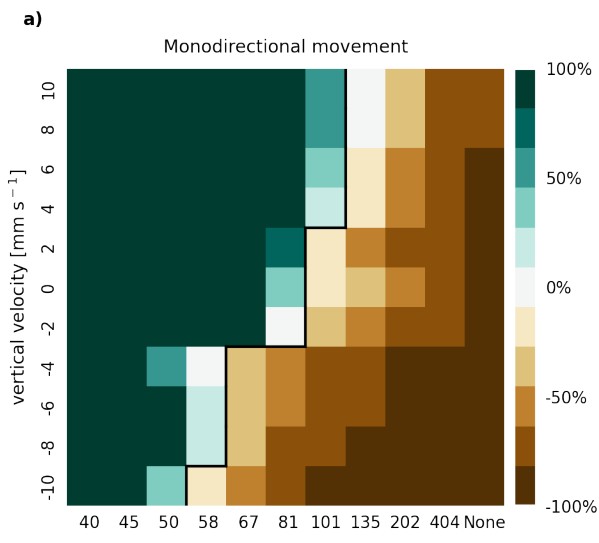

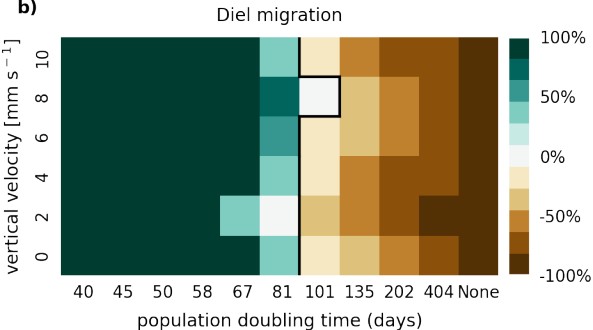

**Figure 3.** Relative population changes for the monodirectional movement (a) and diel migration (b) scenarios. Positive vertical velocities indicate an upwards drift. Positive population changes represent a retention success (green) while negative population changes represent a loss of the population (brown). The vertical black lines indicate the boundary between successfully and unsuccessfully retained scenarios.

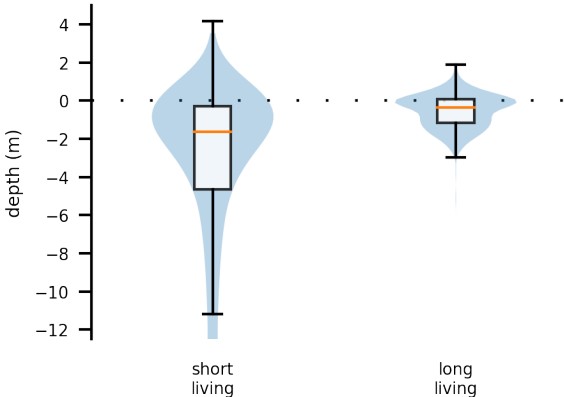

**Figure 4.** Box and violin plot showing the vertical distribution of phytoplankton that are passively drifting. Short-living are those younger than 3 months and long-living all those older than that. Depth is measured relative to the current water surface with positive numbers being above the water surface i.e. stranded on shore.

For the case of the monodirectional movement we see that a higher positive velocity (representing buoyancy) and higher reproduction rates are more beneficial for retention success than a downward oriented velocity (sinking) and lower reproduction rates. As expected, simulations in which the reproduction is set to zero do not show any retention success. While it is easy to understand that high reproduction rates aid retention, we were surprised that buoyant phytoplankton cells are more successful in maintaining their growth in an estuary than sinking ones.

For the case of the diel vertical migration in the velocity range of 4 to 10 $\mathrm{mm\,s^{-1}}$ we see an equal or higher retention success compared to the case with no vertical migration. A diel velocity of 2 $\mathrm{mm\,s^{-1}}$ is less successful than no migration. Most importantly, none of the diel migration scenarios improve the retention success, when compared to passively drifting organisms.

### 3.2 Spatial factors

We are now taking a closer look at spacial factors that allow phytoplankton cells to maintain net growth in the estuary. For this analysis we used data from both sets of experiments i.e. from all cases. Fig. 4 compares two box plots showing the average water depth at the location of each phytoplankton cell between those cells that remained alive for less than three months (short-living) and for more than three months (long-living). Depth is measured relative to the current water surface. Therefore, a value greater than zero indicates that the phytoplankton cell is stranded on the shore during ebb tide. For reference, the water level varies on average by about 5 m due to the tides. (Stanev et al., 2019; Schöl et al., 2014). These analyses show that long-living phytoplankton predominantly live close to the river banks in shallower waters or on tidal flats.

We moreover analyzed the horizontal spacial distribution of long and short-living phytoplankton in fig. 5. To do this, we divide the model domain into equally sized hexagons. The color of each hexagon indicates the average age of the phytoplankton

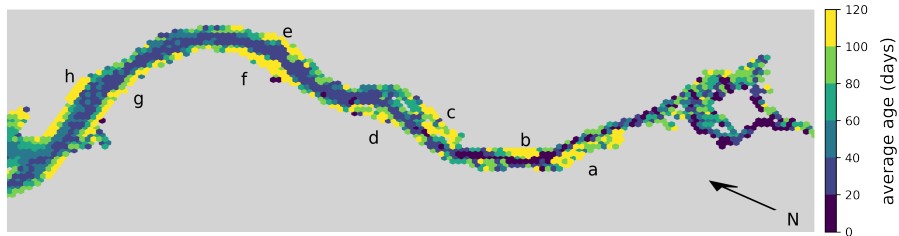

**Figure 5.** Hex-bin heatmap of the average age of phytoplankton cells in the Elbe estuary across all cases. The Hamburgs port area is located on the right with the North Sea to the left. Colors indicate the age of the phytoplankton, with yellowish colors indicating an average age of over three months. Yellow areas are mainly found along the river banks in shallow waters or tidal flats. The important areas are Mühlenberger Loch (a), Wedeler Marsch (b) Haseldorfer Binnenelbe (c), Asseler- and Schwarztonnensand (d), at the mouth of Wischhafener Süderelbe (f), , and Stör (e), and at Nordkedding (g) and Neufelder Marsch (h).

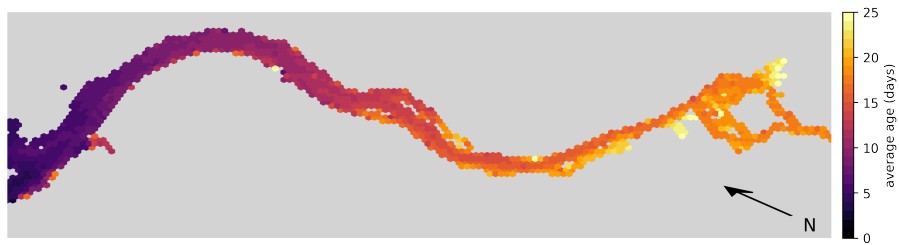

**Figure 6.** Hex-bin heatmap showing average exiting times of the Elbe estuary with Hamburgs port area, as shown in fig. 1, on the right without reproduction, light-limitation, stranding and settling on the riverbed. Colors indicate the time of a water parcel to reach the 20 PSU isohaline from its origin hexagon.

cells within it calculated across all cases. Note, that the spatial age structure is similar for all cases. Hexagons with a yellow
color indicate an average age of over three months. These yellow areas are mainly found along the river banks in shallow waters or tidal flats.

For comparison, the average exit time for water parcels to reach the 20 PSU isohaline per hexagon is shown in fig. 6. This calculation is based on a separate simulation where we released approximately 1.8 million particles homogeneously distributed over the estuary. We released one batch in winter during high discharge conditions on the first of January and another batch
in summer during low discharge conditions at the first of July. Note, that for this simulation reproduction, light-limitation, stranding and settling on the riverbed is disabled to isolate the effect of advection and dispersion.

To further investigate the reasons for the positive effect of buoyancy and the importance of shallow waters and tidal flats, we repeated the first set of simulations and disabled the reproduction of settled and stranded phytoplankton. Under these conditions, populations were unable to retain themselves in the estuary, regardless of their vertical velocity and reproduction
rate indicating that tidal flats are essential for the survival of the population.

### 3.3 Interpretation and contextualization of Results

In this study, we investigated different strategies to explain how phytoplankton populations are able to maintain their population size in estuaries while constantly being at risk to be transported into the open ocean.

The limit of population doubling times that we found necessary for the survival of passively drifting plankton is about 4 months (see fig. 3). Doubling times typically realized in natures are of the order of a few days which is two magnitudes small then those that we found necessary in our model (Koch et al., 2004; Wirtz, 2011). The low reproduction rates required for successful retention demonstrate that our model is also meaningful under more realistic environmental conditions, for example if maximum growth rates cannot be reached due to nutrient or temperature limitations.

Our results suggest that shallow areas are very important for maintaining the estuary phytoplankton population. Plankton that consistently finds itself in areas that fall dry due to the tides will regularly become stranded and therefore not move for much of the tidal cycle. We further see that positively buoyant plankton are more successful in retaining themselves. This is probably because they are more likely to be transported high up on the river bank where the water is less likely to reach them. This effect is emphasized in flatter regions as the distance between the wash margin and constantly flooded areas is larger, increasing the chance of settlement or them stranding again.

Initially, we expected sinking phytoplankton to have a higher retention success than buoyant ones. However, we found that faster sinking phytoplankton are less successful in retaining themselves. Sinking velocities of less than $2\mathrm{mm\,s^{-1}}$ are common for diatoms (Passow, 1991) while larger velocities have been observed for aggregates in the Elbe estuary (Fennessy and Dyer, 1996). Sinking phytoplankton have a reduced downstream velocity because find themselves either settled on the riverbed not moving at all or close to the bed where the average downstream velocity is lower. In addition, the deeper water layers of the Elbe have on average a lower downstream velocity than the upper water column or move upstream due to temperature-induced density stratification (Pein et al., 2021). Nevertheless, buoyant phytoplankton were more successful in their retention in our simulations. The low chance of survival in the estuary for sinking phytoplankton might be explained by their light limitation in deeper waters. We expected phytoplankton to die if they are exposed to dark conditions for more than two weeks. Thus, sinking phytoplankton have a disadvantage to buoyant phytoplankton since they are more likely to become light limited and eventually die. This suggests that dredging has a negative impact on sinking plankton because it increases both depth and turbidity (de Jonge et al., 2014), which increases the aphotic depth and therefore the volume of dark water relative to the volume of illuminated water.

We suspect that the reason for the increased retention success of diel migrating organisms is similar to the monodirectional case. When the upwards diel migration coincides with high tide, phytoplankton are more likely to be stranded far out on the shore, reducing their risk of being washed out quickly. The higher the upward velocities, the greater the chance of being at the waterline during high tide. However, because they are sinking for half of the day they also tend to be light limited more frequent than positively buoyant phytoplankton. It seems like these favorable and unfavorable processes balance each other out, resulting in a similar retention success as for the monodirectional case.

## 3.4 Model limitations & future perspectives

In this study, we aimed to thoroughly investigate different possible retention mechanisms in a complex Lagrangian model system with a highly resolved bathymetry. Due to this computational and spatial complexity, the complexity of the biological particle properties needed to remain simple to keep computational costs manageable, interpretability high and due to a lack of high resolution validation data.

Our model design does not resolve more complex ecosystem dynamics such as nutrient limitation and grazing by higher trophic levels. The Lagrangian model is performed offline, meaning it is not coupled to the Eulerian model that calculates the hydrodynamics and is performed after the fact. Therefore, modeling the advection and dispersal of changes in concentration fields e.g. nutrients due to growth or remineralization was not easily possible. Future modeling efforts could couple the Lagrangian model to a Eulerian model that disperses changes in concentrations fields by biotic activity throughout the model domain. However, this would have drastically increased both, developing and computational time to a point where it would have been infeasible in our time frame and due to the lack of appropriate validation data. The key draw back of this is that growth rates could only be modeled as a constant rate in the current model description, similar to "ad libitum" experiments. This can lead to systematic errors in estimating population growth. In nature, phytoplankton growth is often limited by nutrient availability that nutrient limitation, which slows down the growth of the population, can occur, especially in the most light-saturated areas near the shore. For this reason, we may overestimate the role of shallow areas in our model.

To be consistent with the complexity of the representation of biotic mechanisms, we use a simplistic light limitation. Phytoplankton are expected to be light limited below a water depth of $1$ m and not light limited above this threshold. We have not included a more complex light limitation model that takes into account current light availability and attenuation. A more realistic formulation of light limitation could particularly favor phytoplankton that exhibit diel vertical migration.

A process we mostly ignore in our study is dormancy. Our organisms can survive for $14$ days in light limited waters. However, phytoplankton species have life stages in which they can remain dormant for a longer period of time and germinate again when they find themselves in more favorable waters (Thomas Anderson, 1998). In the process of choosing the light limitation threshold, we conducted sensitivity studies testing the effect of higher light budgets. We found that light budgets over 3 months begin to significantly increase the survivability of sinking organisms, when we crudely assume that they could still reproduce under these conditions. Whether dormancy plays a significant role in an environment where the river bed is continuously dredged is unknown.

Another limitation in our modelling efforts is the lack of sub-grid-resolution structure on the shores. In our representation we assume perfectly flat surfaces with a median distance between nodes of approximately $60$ m. This 'polished' model representation can lead to an underestimation of the retention success, since the surface area on which phytoplankton organisms can settle is underestimated. In nature, vegetation, rocks or other surface irregularities cause a larger surface area on which the phytoplankton organisms can settle in moist conditions.

Our hydrodynamics data set was limited to the year 2012. Therefore, we were not able to study different release times with the same methodology. While we do not expect the general dynamics to change, future research could examine the effect of

varying discharge throughout the seasons on retention and could address the very long term success (>1 year) of the population, as it affected by inter-annual variability and climate change.

While our model does have a settling and resuspension mechanic based on critical sheer velocities we still assume a static bathymetry with sediments not able to move or bury phytoplankton. This masks potential losses due to phytoplankton being buried but also decreases resuspension times.

    Our results clearly suggest the importance of tidal flats and shallow areas along the river banks for the persistence of primary production in the Elbe estuary. However, their effect can currently not be quantified due to the lack of validation
data. Chlorophyll data with a sufficient temporal and spacial resolution is only gathered in the center of the river. Future monitoring efforts should therefore also include data along the river shores on tidal flats or shore-to-shore to quantify the effect of potential future changes by dredging, diking or restoration attempts.

    Frequently stranded plankton have been shown to be essential to the survival of populations in our model. However, data on their ability to survive under these conditions are scarce. Our results suggest that these conditions may be as important as their
ability to quickly regrow under more favorable conditions, and we suggest further research on plankton survivability when stranded.

    For several decades, the annual average chlorophyll concentration in the Elbe estuary has been decreasing (data available at www.fgg-elbe.de/elbe-datenportal.html or see (Hardenbicker et al., 2014; Schöl et al., 2014)), while upstream concentrations do not show this effect. The reasons for this are not fully understood, but one possible reason is the increase in dredge activity.
This increases the average turbidity and thus the aphotic depth, reducing the volume of water in which phytoplankton can grow. A large fraction of the phytoplankton measured upstream of Hamburg port consists of diatoms (Muylaert and Sabbe, 1999), which typically have negative buoyancy (Passow, 1991), making them particularly susceptible to sinking in light-limited waters. Our finding that sinking phytoplankton have a harder time surviving in the estuary supports this theory.

    Another mechanism that might in part explain the drop in phytoplankton concentration at the bathymetric jump, which
is not yet explored in our model, is the phytoplankton stickiness. Phytoplankton, especially blooming one, has been shown to be sticky due to exudates (Kiørboe and Hansen, 1993; van der Lee, 2000; Dutz et al., 2005). Some phytoplankton also produce transparent exopolymer particles, which increase their stickiness to other particles (Windler et al., 2015; De Brouwer et al., 2005). We suspect that this in combination with higher turbidity induced by dredging results in losses due to plankton aggregates sticking to negatively buoyant suspended matter and subsequently sinking to the ground where they are starved of
light. A future model study could create estimates on phytoplankton losses caused by this effect.

## 4   Conclusions

In this study, we investigated the role of different retention strategies for phytoplankton organisms to persist in an estuarine environment. We showed that stranding in shallow nearshore areas is essential for phytoplankton retention, and that phytoplankton that do not strand are rapidly washed away. Our model simulations suggest that growth rates much lower than those
355 observed in nature may be sufficient for populations to prevent their decline due to outwashing, implying that stranding may

be sufficient to maintain the population. Moreover, buoyancy and strong diel vertical migration enhance retention within the estuary. These results highlight the importance of shallow nearshore areas in maintaining the productivity of estuarine ecosystems. Our results suggest that current state-of-the-art models of estuarine ecosystems may overlook an important process and emphasize the need for informed ecosystem-based management to avoid the degradation of estuarine ecosystems by dredging and diking activities.

*Code and data availability.* Input data can be requested from Johannes Pein (johannes.pein@hereon.de). OceanTracker's source code is available atgithub.com/oceantracker/oceantracker. Model configuration and output is available at doi.org/10.25592/uhhfdm.13235

## Appendix A: Sensitivity analysis "dry-out"

In fig. A1 and A2 we present the results of a sensitivity analysis of the stranding mortality threshold, i.e. drying out, from 1 day and 14 days compared to the 7 days shown in fig. 3 Varying this parameters changes the break-even point of growth and loss slightly as expected. However, no regime shift occurs and the observed trends remain the same.

## Appendix B: Sensitivity analysis "light-limitation"

In fig B1 and B2 we present the results of a sensitivity analysis of the mortality threshold due to light limitation of 7 days and 28 days compared to the 14 days shown in fig. 3. Similar to the stranding mortality threshold, perturbations in this parameter change the break-even point of growth and loss as expected. Reducing the tolerated light deficit to half that observed in laboratory studies (Walter et al., 2017) shows a particularly pronounced effect for sinking phytoplankton cells, which are more frequently light limited. This is most clearly visible in the -10 mm case which shows that the break-even point is reached at a doubling time of below 40 days. Nevertheless, the trends discussed e.g. break even points at doubling times much larges then those observed in nature, the favoring of buoyant cells over sinking cells, and the importance of shallow areas remain the same.

## Appendix C: Salinity

In figure C1 shows a map of average salinity of the Elbe estuary. Salinity is averaged in depth and over the whole year.

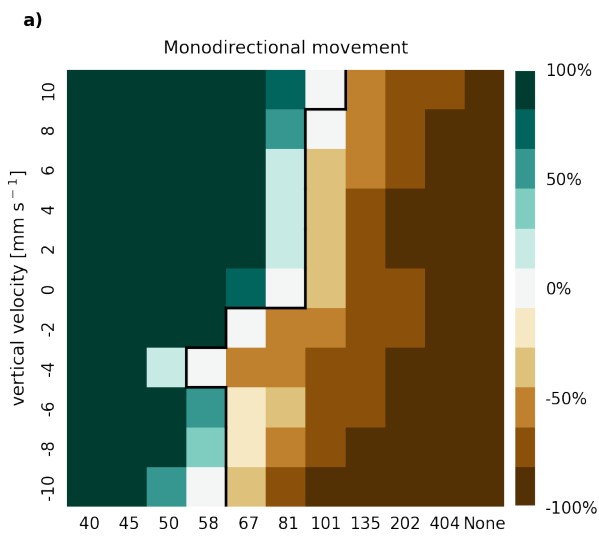

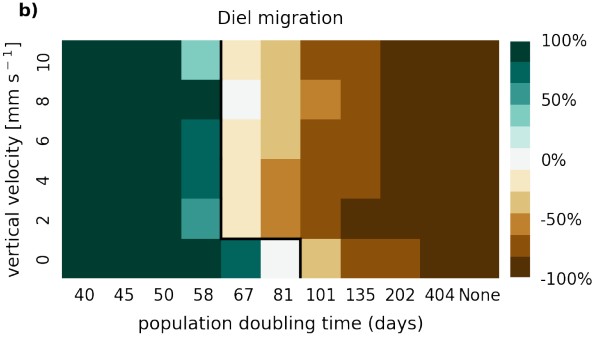

**Figure A1.** Sensitivity analysis for *mortality due to stranding* i.e. drying out showing the retention success similar to fig. 3 with a *threshold of 1 days* without resuspension compared to the 7 days in fig 3 before phytoplankton are culled

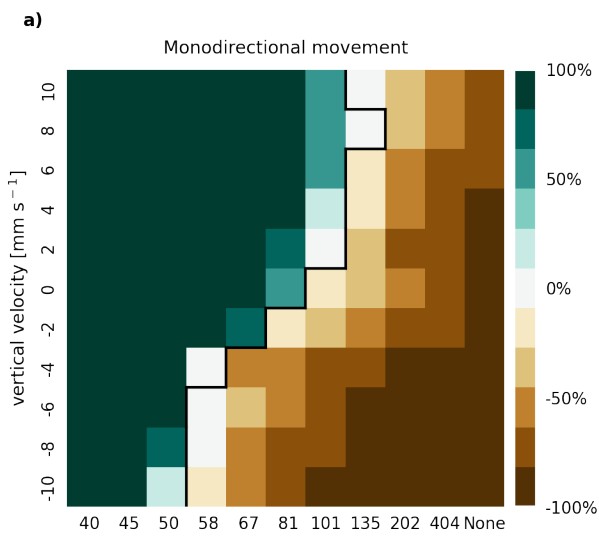

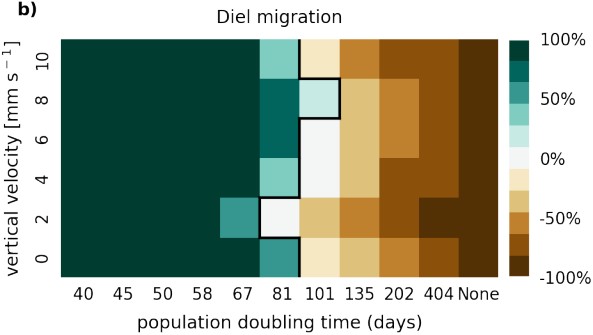

**Figure A2.** Sensitivity analysis for *mortality due to stranding* i.e. drying out showing the retention success similar to fig. 3 with a *threshold of 14 days* without resuspension compared to the 7 days in fig 3 before phytoplankton are culled

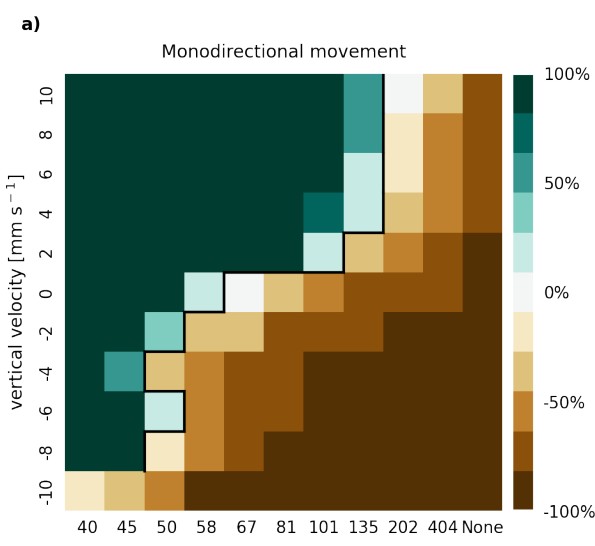

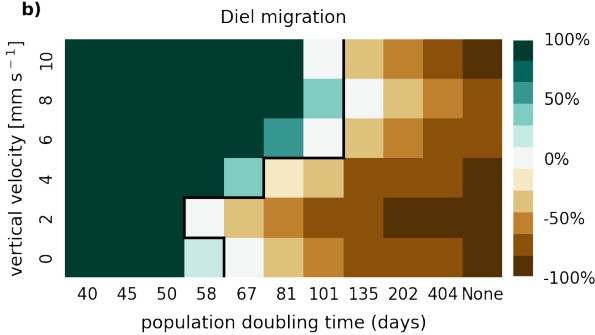

**Figure B1.** Sensitivity analysis for *mortality due to light limitation* showing the retention success similar to fig. 3 with a light deficit *threshold of 7 days* compared to the 14 days in fig 3 before phytoplankton are culled

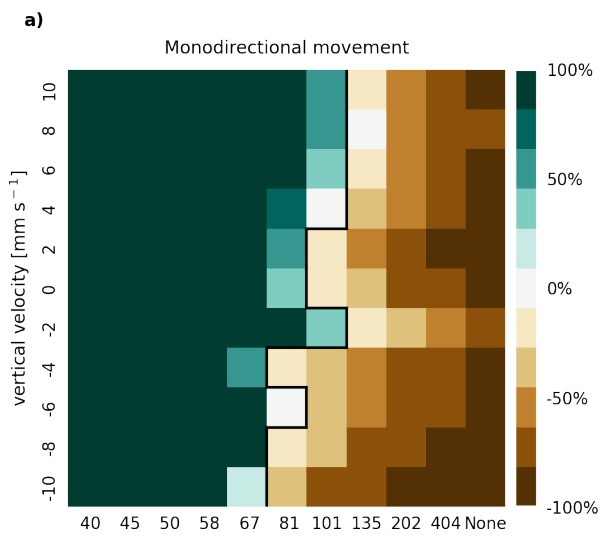

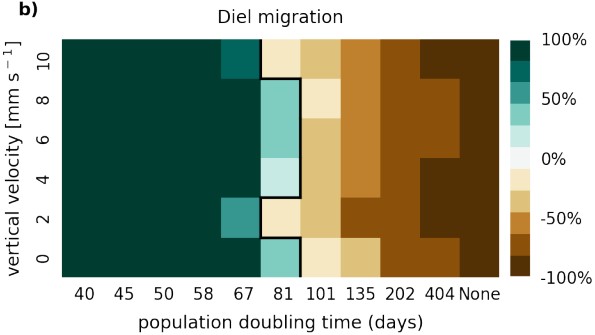

**Figure B2.** Sensitivity analysis for *mortality due to light limitation* showing the retention success similar to fig. 3 with a light deficit *threshold of 28 days* compared to the 14 days in fig 3 before phytoplankton are culled

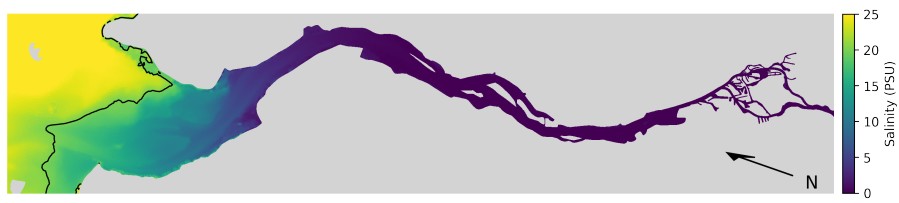

**Figure C1.** Salinity map of the Elbe estuary with Hamburgs port area, as shown in 1, in the bottom right. Salinity is averaged in depth and over the whole year. The 20 PSU isohaline is marked with a black line. Note that this plotted area has been extended downstream compared to fig. 5. Further note that the color map has been capped at 25 PSU for better visibility in low salinity areas.

*Author contributions.* LS and RV contributed to conception of the study. LS designed the studies details and organized the hydrodynamic data. RV provided the source code for OceanTracker. RS, LS improved on the original physical model and LS developed the biological model. LS performed the model simulations, post-processing, and visualization. LS wrote the draft of the manuscript. All authors contributed to manuscript revision, read, and approved the submitted version.


*Competing interests.* The authors declare that they have no conflict of interest.

*Acknowledgements.* We thank Johannes Pein for providing and supporting the implementation of the hydrodynamic data, Jana Hinners for her guidance through the project, and Hans Burchard for his input on dispersion. Further, we thank, Sina Remmers and Philipp Porada for providing helpful comments on the manuscript. This study was funded by the Deutsche Forschungsgemeinschaft (DFG, German Research Foundation) within the Research Training Group 2530: "Biota-mediated effects on Carbon cycling in Estuaries" (project number 407270017), contribution to Universität Hamburg and Leibniz-Institut für Gewässerökologie und Binnenfischerei (IGB) im Forschungsverbund Berlin e.V.

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
