# Peer review of "Phytoplankton Retention Mechanisms in Estuaries: A Case Study of the Elbe Estuary"

_EGUsphere, 2023_

## Author Response (AR1)

**Response to reviewers**

**Response to reviewer RC1**

- **1. Terminology: Instead of referring to 'particles,' using 'phytoplankton cells' would better align with the biological focus of their study.**

  We had a similar discussion when we were writing the paper, so we particularly value this input. In the end, we chose "particle" to emphasise the abstraction that takes place in our model when representing a phytoplankton cell, as our model is of course not able to fully capture their behaviour and dynamics. Ideally, therefore, these two concepts should not be confused by the reader. However, as you suggested, the term "particle" might be more confusing, especially for non-modeling readers, as it obscures the biological implications, and we have changed the term "particle" to "phytoplankton cell" where appropriate, as you suggested.

  E.g. we changed the paragraph introducing the concept of a lagrangian model to read:

  > [...] [Eulerian models] lack temporal consistency, meaning that the life history and trajectory of a phytoplankton cell cannot be tracked. Previous modeling studies have attempted to overcome this problem using a Lagrangian approach. A Lagrangian model does not try to track e.g. concentrations at fixed positions, but rather follows the motion of individual particles that can be used to represent e.g. water parcels or organisms. Their ability to resolve the interactions of individual phytoplankton cells or aggregates with the bathymetry, e.g. through settling or stranding, while maintaining temporal consistency, is essential for investigating retention mechanisms.

- **2. Model Validation: Including aspects related to model validation, such as capturing the seasonal cycle, the duration of bloom events, the spatial distribution of blooms in relation to distance from the estuary, and other relevant parameters, would enhance the biological relevance of their research and provide a more comprehensive understanding of the dynamics at play.**

We agree that model validation is important, but it seems to us that there may be a misunderstanding about the type of model presented, its implications and purpose.

Our model can be interpreted (in part) as a post-processing or further analysis of the model presented by Pein et al. (2021). The model of Pein et al. is a Eulerian model that captures both hydrodynamics and biology. The biology is modelled using ECOSMO. This model includes several planktonic compartments (diatoms, flagellates, cyanobacteria and two zooplankton compartments) and has been successfully applied in several areas (Schrum et al., (2006), Daewel and Schrum (2013)). In the model of the Elbe estuary that we use, it is calibrated and validated with observational data from both long-term measuring stations and cruises that take transects in the centre of the channel. The Pein et al. model is able to predict population dynamics at the concentration level reasonably well. It captures both the seasonal cycle, i.e. the bloom events, and the spatial distribution.

Our model does not attempt to predict population dynamics. We use our model to shed light on physical processes that are largely ignored in Eulerian models such as that of Pein et al. (2021) - in particular, the process of stranding and other interactions with the bathymetry. Our focus is therefore, in a sense, to help understand the loss term induced by outwashing to high salinity waters or dry shores. These processes are not directly represented in the differential equation of the Eulerian model that predicts ecosystem dynamics and therefore can not be easily studied with such models. For this purpose, we have chosen a Lagrangian approach, which allows us to model phytoplankton stranding in a simple and (computationally) inexpensive way, with a temporal consistency that is crucial for modelling the processes studied and that cannot be achieved with Eulerian models.

Because of this narrow focus, we have simplified the biological processes as much as possible to allow for high interpretability of our results. We agree that it would be desirable for our model to predict the full ecosystem dynamics, as this would potentially improve the interpretability of the effects of the processes studied, i.e. to make quantitative rather than just qualitative predictions. However, not only would this greatly increase the cost of building, running and evaluating such models, it is not currently possible due to technical constraints and lack of calibration and validation data, neither in our Lagrangian model nor in any other model to our knowledge.

We this paragraph to emphasize the already validated hydrodynamical and ecostsystem model on which our study is based:

> Nevertheless, there are sophisticated estuarine models that are able to reproduce the complex dynamics of estuaries reasonably well. This includes currents and water levels on the

physical side, but also chlorophyll concentrations and other biologically driven properties (Pein et al. (2021), Schoel et al (2014)). However, these are Eulerian models. This means that they are based on a fixed grid and calculate the concentration of a tracer, such as phytoplankton, at each grid cell. This makes it difficult to study concepts such as retention times, as they lack temporal consistency, meaning that the life history and trajectory of a phytoplankton cell cannot be tracked. Previous modeling studies have attempted to overcome this problem using a Lagrangian approach. A Lagrangian model does not try to track e.g. concentrations at fixed positions, but rather follows the motion of individual particles that can be used to represent e.g. water parcels or organisms. The ability to resolve the interactions of individual phytoplankton cells or aggregates with the bathymetry, e.g. through settling or stranding, while maintaining temporal consistency, is essential for investigating retention mechanisms.

and the following section in the "model limitations" section:

In this study, we aimed to thoroughly investigate different possible retention mechanisms in a complex Lagrangian model system with a highly resolved bathymetry. Due to this computational and spatial complexity, the complexity of the biological particle properties needed to remain simple to keep computational cost manageable and due to a lack of high resolution validation data.

Our model design does not resolve more complex ecosystem dynamics such as nutrient limitation and grazing by higher trophic levels. The Lagrangian model is performed offline, meaning it is not coupled to the Eulerian model that calculates the hydrodynamics and is performed after the fact. Therefore, modeling the advection and dispersal of changes in concentration fields e.g. nutrients due to growth or remineralization was not easily possible. Future modeling efforts could couple the Lagrangian model to a Eulerian model that disperses changes in concentrations fields by biotic activity throughout the model domain. [...]

- **3. The aspect that phytoplankton cells survive in the dry grid cell (without water) needs to be justified. Otherwise, this should be corrected in the method, post-processing (without redoing all the tests) these cells can be excluded from the final count, and further conclusions should be corrected.**

We agree that pythoplankon cannot survive indefinitely in dry conditions. To contextualise their ability to survive, we would like to highlight two things:

First, the areas where phytoplankton strand are typically frequently flooded by the tide. The vast majority of phytoplankton in our model are stranded for less than one tidal cycle, i.e. less than 12 hours. Secondly, cells that are considered 'dry' by the model are not necessarily devoid of water. The cell resolution in these areas is typically between 50 and 100m. Cells are considered dry if the water level falls below 0.1m over the majority of their area. Therefore, a lot of sub-resolution structure can be expected. These include sand ripples, tidal creeks or small pools that hold water where phytoplankton could survive for several days before drying out. In addition, the low marsh that surrounds most of the estuary contains a lot of vegetation, typically tall reeds. This is thought to improve the survivability of the phytoplankton around it by increasing soil moisture long enough for most cells to survive through a tidal cycle.

The stranding and resuspension of phytoplankton and microphytobenthos has been shown to be an important process for primary production under eastuarine conditions (Carlson et al (1984), De Jonge et al (1992), Kromkamp et al (1995), Savelli et al (2019)). While this process is particularly well established for microphytobenthos (Savelli et al), Semcheski et al (2016) showed that the distinction between 'phyoplankton' and 'microphytobenthos' is fuzzy with a large overlap.

To our knowledge, no study has investigated the survival of stranded phytoplankton under estuarine conditions. We therefore tested a range of parameter choices before publication and have now added a sensitivity analysis in the appendix to show that time to dry-out is not a particularly sensitive parameter. Testing parameters from 1 to 30 days showed no regime shift in our results. We chose the 7 day cut-off because we felt it was a reasonable time frame under the conditions observed in the tidal marshes, and there were no observational data to suggest a better choice.

We have also added the following first paragraphs and adjusted the second in the methods section to better contextualise this choice for the reader:

> We consider phytoplankton cells that are stranded out of the water by the receding tide, and lie dry for more than 7 consecutive days to be dead and remove them. Note that these dry cells are not necessarily completely devoid of water, but are considered dry if the majority of its area has a water level below 0.1 m. Additionally, in nature these areas typically contain small sub-resolution structures like tidal ripples or small puddles and vegetation. There are currently no studies investigating the time range for survival of stranded phytoplankton on tidal-flats or marshes in estuaries. Therefore, we performed a sensitivity analysis to determine the effect of this parameter on the retention success of the phytoplankton population (see appendix section A).
> We include a settling and resuspension model to represent

tidal stranding and phytoplankton cells settling on the bed of the estuary. Stranding phytoplankton and microphytobenthos have been shown on several occasions to be a major driver of estuarine primary production (Carlson et al., 1984; De Jonge and Van Beuselom, 1992; Kromkamp et al., 1995; Savelli et al., 2019). Phytoplankton cells become stranded when the current grid cell becomes dry and stay in place until they are resuspended or dry-out. They are not allowed to move from wet cells to dry cells, by the random walk diffusion applied to all phytoplankton cells. A grid cell is considered dry based on the flag given in the SCHISM hydrodynamic model output. Once this cell is flooded again, all the stranded phytoplankton cells are resuspended and able to move again.

- **Page 2, lines 30-35: It appears that the author may be conflating two distinct diel migration behaviors observed in planktonic species. One type of diel migration is exhibited by phytoplankton, which is primarily driven by the availability of sunlight for photosynthesis. This behavior is solely dependent on the sun's position in the sky, as phytoplankton are primary producers that rely on light for their metabolic processes. On the other hand, carnivorous planktonic species, like certain zooplankton and dinoflagellates, exhibit a different diel migration pattern. Their vertical movements are not directly driven by the sun but are instead motivated by the distribution of their prey, mainly phytoplankton, which, in turn, is influenced by sunlight-driven photosynthesis. These species engage in diel migration as a survival strategy, often to avoid predators or to exploit variations in food availability. In this context, it is essential to emphasize the distinction between these two types of diel migration patterns to provide a more accurate and biologically informed account of the behaviors of planktonic organisms. Recognizing the ecological drivers behind these migrations is crucial for a comprehensive understanding of aquatic ecosystems.**

We agree that the reason why diel migration is beneficial for autotrophs, mixotrophs and heterotrophs is different. As we study phytoplankton, we focus on autotrophic and mixotrophic plankton. Therefore, all model organisms benefit from diel migration by maximising light capture while potentially avoiding grazing, while the mixotrophs may additionally benefit by following food or nutrient sources. In all cases, however, the consequence remains the same: an upward movement during the day and a downward movement at night.

While there may be two reasons for the diurnal migration, whatever the cause, the purpose of this paper is to examine the effect of this migration on retention.

We changed the mentioned paragraph to make this clearer. It now reads:

> Diel vertical migration is a process where organisms move up and down in the water column in response to the sun. This movement may favors retention by allowing plankton to reduce the time in the faster downstream currents at the water surface. A study by Anderson and Stolzenbach (1985) showed that diel migrating dinoflagellates were able to out compete other non-motile phytoplankton in an embayment environment and even compensate for outwashing losses through reproduction increasing their abundance. However, this also implies that the growing part of the population is somehow retaining their position. If the regrowing population is also continuously drifting downstream they will not able to sustain their population in that area and ultimately die out due to unfavorable salinity conditions in marine waters (Admiraal, 1976; von Alvensleben et al., 2016; Jiang et al., 2020). The presence of diel migration has mostly been demonstrated for motile phytoplankton such as dinoflagellates (Hall et al., 2015; Crawford and Purdie, 1991; Hall and Paerl, 2011) and zooplankton species (Kimmerer et al., 2002). While the motivation for diel migration for autotrophic, mixotrophic, and heterotrophic differs, the consequence remains the same, an upward movement during the day and a downward movement during the night.

- **Page 4, lines 83-84: What is the spatial resolution of the three-dimensional unstructured grid used to represent the Elbe estuary in this model, and how does it vary within the dataset?**

We added more detailed information on the gridding of the model domain in the methods sections as requested. It now reads:

> The unstructured mesh is three-dimensional and consists of 32k nodes using terrain-following coordinates based on the LSC2 technique (Zhang et al., 2016) for the vertical grid, allowing a maximum number of 20 levels. Regions with depths less than 2 m are resolved by only one vertical level. The bathymetric data were provided by the German Federal Maritime and Hydrographic Agency (Bundesamt fuer Seeschifffahrt und Hydrographie, BSH) and the German Waterways Agency (Wasserstraßen- und Schiffahrtsamt, WSA) with a horizontal resolution of 50 m in the German Bight, 10 m in the Elbe estuary and 5 m in the Hamburg port [?]. [...] The model provides us with a node-based mesh containing a range of information [...] and a dynamically varying spacial resolution with distance between nodes ranging from 5 to 1400 m with a median distance of approximately 75 m

- **Page 5 lines 107-110: The statement, "A particle starts its life with a light budget of 28 days, and each minute below 1m reduces this budget by one minute, while the opposite applies when they are above 1m. Children of light-limited parents inherit the remaining light budget of their parents," should be supported by relevant laboratory studies or evidence. Additionally, the terminology used, such as "children" and "parents" for phytoplankton, might be confusing and should be rephrased for clarity.**

We changed the paragraph as suggested to better explain the choice and avoiding the term "children" and "parents". We also fixed a typo incorrectly stating the light budget used in the model in this section. The mentioned paragraph now reads:

> Phytoplankton cells will also die if they are light-limited for 14 days. This value is based on measurements presented in (Walter et al., 2017) which imply that the majority of phytoplankton is dead after 14 days of light limitation. A sensitivity analysis for this parameter is presented in sec. B suggesting no strong influence on the retention success. They are considered light-limited below a depth of 1m based on SPM data presented in (Stanev et al., 2019). The initial batch of phytoplankton cells starts their life with a full light budget of 14 days, and each minute below 1m reduces this budget by one minute, while the opposite applies if they are above 1m. When a cell splits both inherit the same remaining light budget.

- **Page 5 lines 118-122: The statement that "particles become stranded when the current grid cell becomes dry, and once this cell is rewetted, all stranded particles resuspend and are able to move again" should be justified based on ecological principles and the behavior of phytoplankton. It's important to explain the reasoning behind this choice, as phytoplankton typically cannot survive when completely dry.**

We justified this choice in our answer to 3.) above as requested. In short grid cells are typically not completely dry and phytoplankton cells typically rewettet in less then 12 hours. We added a paragraph in the paper to reflect our arguments and updated the mentioned paragraph as also presented in our response to 3.). It now reads:

> We consider phytoplankton cells that are stranded out of the water by the receding tide, and lie dry for more than 7 consecutive days to be dead and remove them. Note that these dry cells are not necessarily completely devoid of water, but are considered dry if the majority of its area has a water level below 0.1 m. Additionally, in nature these areas typically contain small

sub-resolution structures like tidal ripples or small puddles and vegetation. There are currently no studies investigating the time range for survival of stranded phytoplankton on tidal-flats or marshes in andTherefore, we performed a sensitivity analysis to determine the effect of this parameter on the retention success of the phytoplankton population (see appendix section A).

We include a settling and resuspension model to represent tidal stranding and phytoplankton cells settling on the bed of the estuary. Stranding phytoplankton and microphytobenthos have been shown on several occasions to be a major driver of estuarine primary production (Carlson et al., 1984; De Jonge and Van Beuselom, 1992; Kromkamp et al., 1995; Savelli et al., 2019). Phytoplankton cells become stranded when the current grid cell becomes dry and stay in place until they are resuspended or dry-out. They are not allowed to move from wet cells to dry cells, by the random walk diffusion applied to all phytoplankton cells. A grid cell is considered dry based on the flag given in the SCHISM hydrodynamic model output. Once this cell is flooded again, all the stranded phytoplankton cells are resuspended and able to move again.

- **Page 6, line 150: Please provide an explanation for the choice of population doubling times in idealized conditions ranging from 40 to 404 days. This choice should be based on scientific rationale and may require further clarification.**

Under ideal conditions, phytoplankton doubling times are much lower than the range tested in our model, with doubling times of less than one day. These ideal cases are of course rare, as phytoplankton are almost always strongly limited in nature, e.g. by light or nutrient availability.

In our study we are examining the impact of a range of physical drivers, most importantly losses due to outwashing of phytoplankton and are trying to decouple the biological drivers as much as possible to achieve a better interpretability of the results. Hence, we chose our doubling times not to accuratley represent fission rates observed in nature but such that they allow us to estimate the losses due to physical drivers, which in our case are light limitation, outwashing to the shores and to the sea. The presented doubling times in our study can be interpreted as potential average net-doubling-times in the presence of predation and mortality, nutrient availability. We are not trying to representing the ecosystem dynamics by natural growth and mortality of phytoplankton as this is already done in the cited study Pein et al. (2021) where they include a full ecosystem model but lack the possibility to represent the process (e.g. stranding) simulated and studied here.

We added a comment to clarify this to the mention paragraph. It now reads:

Each vertical velocity is examined for a range of different reproduction rates, expressed as population doubling times ranging from 40 to 404 days with a logarithmic scaling. In the following, we will use reproduction rate to refer to the prescribed population growth rate under idealized conditions and use growth rate whenever we describe population growth in nature. The prescribed population growth rate can be interpreted as potential average net-doubling-times in the presence of predation and mortality, nutrient availability while testing the effect of out-washing.

- **Page 7, Section "Results": Before analyzing the retention success, it's advisable to perform some form of model validation. Consider whether your model or specific scenarios with their parameters successfully reproduce the seasonal cycle of phytoplankton, including the duration of bloom events and the number of particles over distance from the North Sea. Model validation is crucial to ensure the reliability of your results.**

This request is similar to point 2.) where we explained why this model does not attempt to predict population dynamics. We agree that model validation is important to ensure the reliability of model results, which is why we use the hydrodynamics of an ecosystem model with validated population dynamics. However, to our knowledge, no observational studies have been conducted to investigate the mechanism of phytoplankton retention under estuarine conditions and spatial distribution at finer scales. In fact, the lack of field studies was the main motivation for this modelling study, as we try to emphasise the importance of these processes and suggest that such experiments should be carried out. Quantifying the importance of these processes in the field is essential before they can be added to the current state of the art models to better represent phytoplankton losses, which are currently fitted to observational data mainly using natural mortality and grazing parameters.

We have added the following paragraph, as previously stated in our response to 2.) above. We added this paragraph to emphasise the already validated hydrodynamic and ecosystem model on which our study is based:

[...] there are sophisticated estuarine models that are able to reproduce the complex dynamics of estuaries reasonably well. This includes currents and water levels on the physical side, but also chlorophyll concentrations and other biologically driven properties (Pein et al. (2021), Schoel et al (2014)). However, these are Eulerian models. This means that they are based on a fixed grid and calculate the concentration of a tracer, such as phytoplankton, at each grid cell. This makes it difficult to study concepts such as retention times, as they lack temporal

consistency, meaning that the life history and trajectory of a phytoplankton cell cannot be tracked. [...]

and the following section in the "model limitations" section:

> In this study, we aimed to thoroughly investigate different possible retention mechanisms in a complex Lagrangian model system with a highly resolved bathymetry. Due to this computational and spatial complexity, the complexity of the biological particle properties needed to remain simple to keep computational cost manageable and due to a lack of high resolution validation data.
>
> Our model design does not resolve more complex ecosystem dynamics such as nutrient limitation and grazing by higher trophic levels. The Lagrangian model is performed offline, meaning it is not coupled to the Eulerian model that calculates the hydrodynamics and is performed after the fact. Therefore, modeling the advection and dispersal of changes in concentration fields e.g. nutrients due to growth or remineralization was not easily possible. Future modeling efforts could couple the Lagrangian model to a Eulerian model that disperses changes in concentrations fields by biotic activity throughout the model domain. [...]

And further emphasised the point that this study suggest and shall work as a foundation for future field measurements in the outlook. It now reads:

> Our results clearly suggest the importance of tidal flats and shallow areas along the river banks for the persistence of primary production in the Elbe estuary. However, their effect can currently not be quantified due to the lack of validation data. Chlorophyll data with a sufficient temporal and spacial resolution is only gathered in the center of the river. Future monitoring efforts should therefore also include data along the river shores on tidal flats or shore-to-shore to quantify the effect of potential future changes by dredging, diking or restoration attempts. Frequently stranded plankton have been shown to be essential to the survival of populations in our model. However, data on their ability to survive under these conditions are scarce. Our results suggest that these conditions may be as important as their ability to quickly regrow under more favorable conditions, and we suggest further research on plankton survivability when stranded.

- **Page 7, line 171: Please clarify the intention behind looking at the state of phytoplankton after one year in terms of estimating areas where they "successfully retain."**

This is a reference to our 'retention metric' defined on line 158ff. Conceptually, we consider a population to be successfully maintained if it shows long-term growth. We consider one year to be a reasonable "long-term" time frame for this, firstly because it is much longer than the typical outwash period (see newly added Figure 6) of up to 3 weeks, and secondly because it represents all the major seasonal cycles, in particular the upstream seasonal runoff cycle and the downstream seasonal and tidal cycles.

We have modified the "retention metric" paragraph as you suggested to reflect the reasoning presented here. It now reads:

> Conceptually, we consider a population to be successfully retained if it is able to sustain itself long term or even shows growth. Practically, this is evaluated by comparing the population size at the end of the year to the size after release. The choice of one year is considered reasonable because it covers the full seasonal cycle and is also much longer than the average exit or flushing time of the estuary (see fig. 6).

and added a paragraph to the outlook:

> Our hydrodynamics data set was limited to the year 2012. Therefore, we were not able to study different release times with the same methodology. While we do not expect the general dynamics to change, future research could examine the effect of varying discharge throughout the seasons on retention and could address the very long term success (¿1 year) of the population, as it affected by inter-annual variability and climate change.

- **Page 7, line 177: When stating "approximately 3 months," consider providing supporting evidence or references to confirm the accuracy of this time frame based on relevant observations or studies.**

This is not based on other studies but is a reference to our results presented in fig. 3 where the break even point between physically induced loses and growth lies in between 81-101 days. which are approximatly 3 months. The mentioned paragraph now references this:

> Our simulations show that the population is able to successfully retains itself under certain conditions. Passively drifting phytoplankton is able to sustain themselves in the estuary if they have a reproduction rate that doubles their population size within approximately 3 months (see fig. 3)

- **Page 9, Figure 4: The positive depths shown in Figure 4 may be related to tidal oscillations. It would be valuable to describe the tide variabilities or free surface level variability in the site section to help explain these depth variations.**

Yes, the positive values are caused by tidal oscillations that lift phytoplankton cells into areas where they become stranded during ebb tides. We have added a contextualising comment to the tidal range to make this clearer to the reader:

> Fig. 4 compares two box plots showing the average water depth at the location of each phytoplankton cell between those cells that remained alive for less than three months (short-living) and for more than three months (long-living). Depth is measured relative to the current water surface. Therefore, a value greater than zero indicates that the phytoplankton cell is stranded on the shore during ebb tide. For reference, the water level varies on average by about 5 m due to the tides. (Stanev et al., 2019; Schöl et al., 2014).

- **Page 9, line 196: Please clarify which tests or scenarios were chosen to be plotted on Figure 5. Explain whether this is an average over all the tests conducted and provide justification for this choice.**

Since we have no reason to favour or emphasise any particular case, we use an average calculated over all cases, or tests as you call them here. Furthermore, all cases follow the same spatial pattern when plotted individually, with no significant shift in the structure of the average age map, as they all rely on stranding processes to retain themselves, as shown in Figure 4.

In order to make this clear to the reader, we have modified the referenced paragraph, which now reads

> We moreover analyzed the horizontal spacial distribution of long and short- living phytoplankton in fig. 5. To do this, we divide the model domain into equally sized hexagons. The color of each hexagon indicates the average age of the phytoplankton cells within it calculated across all cases. Note, that the spatial age structure is similar for all cases. Hexagons with a yellow color indicate an average age of over three months. These yellow areas are mainly found along the river banks in shallow waters or tidal flats.

- **Page 9, line 195: The statement regarding the parameterization of drifting particles as phytoplankton and their tendency to strand near riverbanks should be approached with caution. Phytoplankton typically cannot survive away from water. To provide a more accurate assessment of phytoplankton behavior, consider excluding particles that become stranded in dry grid cells and correlating their behavior with currents over the coasts**

**and tides, as these factors are usually lower near the coasts, favoring retention.**

As discussed in our response to point 3), we do indeed remove particles/phytoplankton aggregates that become stranded after a period of time, and cells flagged as 'dry' have a lot of sub-resolution structure. Perhaps a better name for this scihsm flag would have been 'not-flooded' as these cells are in most cases quite moist.

Regarding your comment after "To provide [...]", we are not quite sure what you are referring to. If you are asking how the currents are calculated and how they affect the movement of the phytoplankton: The trajectory of a phytoplankton is driven by currents and tides. They move almost instantaneously with the currents and follow them, ignoring diffusion for the moment.

They are therefore correlated. Advection and diffusion were calculated by Pein et al. (2021) using SCHISM solving the Navier-Stokes equations, and their behaviour, which in our case is a kind of vertical motion, is added by us. This implies that we represent the currents and tides along the coast as accurately as possible in our model resolution. This is discussed in more detail in the Pein paper, where the validation process with tides and currents is shown. The currents in shallow water are also slower in our model than in deeper water, as you suggested. This is mainly due to friction between the water layer and the river or sea bed.

Alternativly, if you are referring to the inclusion of a dynamic behaviour: During the early conceptual development of this model we also considered including a vertical migration process of the phytoplankton depending on their velocity, e.g. that they move up and down depending on their speed relative to the coast.

However, we couldn't find any observations showing that pytoplankton exhibit such a migration behaviour or any other behaviour that would suggest that they are somehow able to feel their speed, only their acceleration. One could consider acceleration as a driver for migratory behaviour. However, we did not find any study showing that phytoplankton also exhibit acceleration-dependent migratory behaviour either. We therefore decided to include only light-dependent migration, i.e. moving up and down with the sun.

A phytoplankton cell is moved by three processes: Advection (currents influenced by tides), Diffusion (e.g. turbulence) and their behavior.

- **Page 10, Figure 5: If possible, mark the important sites labeled as "a," "b," "c," etc., on Figure 1 to provide a clearer reference for readers.**

The areas labelled in Figure 5 are not visible in Figure 1. Figure 1 only shows the port area, which is the far right part of Figure 5. We have added

a comment to the figure description to help the reader to align these to maps.

**Page 10, lines 210-220: Please cite relevant observations or studies where phytoplankton survival without water is documented to support the statement made in this section. If it cannot be supported, all conclusions about retention in tidal flats should be rewritten.**

We added citations to relevant observations or studies where phytoplankton survival without water is documented to support the statement made in this section and as described in our response to 3.)

In short they are:

> [...]. Stranding phytoplankton and microphytobenthos have been shown on several occasions to be a major driver of estuarine primary production (Carlson et al., 1984; De Jonge and Van Beuselom, 1992; Kromkamp et al., 1995; Savelli et al.,2019).

- **Conclusion section is very poor and need to be revised.**

We reworked the conlusion as suggested and it now reads:

> In this study, we investigated the role of different retention strategies for phytoplankton organisms to persist in an estuarine environment. We showed that stranding in shallow nearshore areas is essential for phytoplankton retention, and that phytoplankton that do not strand are rapidly washed away. Our model simulations suggest that growth rates much lower than those observed in nature may be sufficient for populations to prevent their decline due to outwashing, implying that stranding may be sufficient to maintain the population. Moreover, buoyancy and strong diel vertical migration enhance retention within the estuary. These results highlight the importance of shallow nearshore areas in maintaining the productivity of estuarine ecosys- tems. Our results suggest that current state-of-the-art models of estuarine ecosystems may overlook an important process and emphasize the need for informed ecosystem-based management to avoid the degradation of estuarine ecosystems by dredging and diking activities.

**Response to reviewer RC2**

- **On the numerical velocity fields: boundary conditions, resolution, vertical components, explicit bathymetry, etc...**

We added the requested information to the description of the hydrological in the introduction. It now reads:

We use the hydrodynamic data generated by the latest SCHISM model of the Elbe estuary (Pein et al., 2021) from the weir at Geesthacht to the North Sea, including several side channels and the port area (see figure 2). SCHISM solves the Reynolds-averaged Navier-Stokes equations on unstructured meshes assuming hydrostatic conditions with a time step of 60 s. The unstructured mesh is three-dimensional and consists of 32k nodes using terrain-following coordinates based on the LSC2 technique (Zhang et al., 2016) for the vertical grid, allowing a maximum number of 20 levels. Regions with depths less than 2 m are resolved by only one vertical level. The bathymetric data were provided by the German Federal Maritime and Hydrographic Agency (Bundesamt fuer Seeschifffahrt und Hydrographie, BSH) and the German Waterways Agency (Wasserstraßen- und Schiffahrtsamt, WSA) with a horizontal resolution of 50 m in the German Bight, 10 m in the Elbe estuary and 5 m in the Hamburg port Stanev et al. (2019). The boundary conditions on the seaward side include sea surface elevation, horizontal currents, salinity and temperature Stanev et al. (2019) and discharge and temperature from the Elbe river on the landward side. Atmospheric forcing includes wind, air temperature, precipitation, shortwave and longwave radiation Stanev et al. (2019). Model validation is based on tide gauge stations and long-term stationary measurements of salinity, water temperature, and horizontal currents. Biochemical variables, including chlorophyll, are based on long-term measurements at the Seemannshöft and Grauerort stations Pein et al. (2021). The model provides us with a node-based mesh containing a range of information such as water velocity, salinity, water level and dispersion. The year represented in that dataset is 2012 with a temporal resolution of 1 hour and a dynamically varying spacial resolution with distance between nodes ranging from 5 to 1400 m with a median distance of approximately 75 m.

- **On the Lagrangian transport model: interpolation schemes, use of eddy diffusion?, sticking of particles to land, etc... You may also compute other Lagrangian quantities to show like exit times, retention times, or accumulation zones (vertical and horizontal).**

We modified the section describing the interpolation scheme, use of eddy diffusion and how they stick to the land to be clearer. Thank you very much for the suggestion to include "exit-times". We added a figure showing average exit times in the results.

The section describing eddy diffusivity now reads:

Phytoplankton cells are not only advected but also diffused

based on eddy diffusivity which is crucial to represent tidal-pumping processes. Diffusion was modeled using a random walk using a random number generator with a normal distribution. Horizontally the standard distribution of the random walk is set to 0.1 m/s. The vertical displacement of a phytoplankton cell $\partial z\ i$ is calculated by

$$\partial z_i = K_v^{'}(z_i(n))\partial t + N(0, 2K_v(z_i)) \tag{1}$$

based on Yamazaki et al. (2014) where $z_i$ is the vertical position of the phytoplankton cell, $K_v^{'}$ is the vertical eddy diffusivity gradient, $K_v$ is the vertical eddy diffusivity and $N$ is the normal distribution. The term based $K_v^{'}$ is needed to avoid phytoplankton accumulation on the top and bottom of the water column from the hydrodynamic model output.

The section describing the interpolation scheme now reads:

Flow velocities, like any other hydrodynamic data, were interpolated linearly in time, linearly in space on the vertical axis and on the horizontal axis using barycentric coordinates, with175 the exception of water velocity in the bottom model cell, where logarithmic vertical interpolation is used.

The section describing sticking to land now reads:

Phytoplankton cells become stranded when the current grid cell becomes dry and stay in place until they are resuspended or dry-out. They are not allowed to move from wet cells to dry cells, by the random walk diffusion applied to all phytoplankton cells. A grid cell is considered dry based on the flag given in the SCHISM hydrodynamic model output. Once this cell is becomes flooded again all stranded phytoplankton cells resuspend and are able to move again

- **On the "population dynamics": how particles divide, die (is it a Gillespie simulation)? number of particles, density, etc...**

We do not use a Gillespie simulation. Phytoplankton cells die independent off the presence of other cell but only based on enviromental conditions. Hence we do not need to enforce a mass balance as required in a Gillespie simulation. We updated the paragraph describing the particle division, mortality and the paragraph describing particle density.

The section regarding reproduction now reads:

Reproduction is represented as a fission process, where each phytoplankton cell has a probability to split effectively producing a copy. This is a novel feature [...] We perform multiple

simulations for a range of reproduction rates, implemented as a fission probability evaluated every minute, that are constant over the lifetime of the cell. While a fixed reproduction rate is a simplification that does not allow for more realistic simulation115 of the population dynamics of a particular species, it does allow us to investigate the general mechanisms that enable plankton retention.

The section regarding mortality now reads:

Mortality is induced by one of three processes: high salinity, when they dry out while stranded, or due to long term light limitation. When particles cells are exposed to high salinity water above 20PSU, a mortality probability of 0.5applied removing dead phytoplankton cells from the simulation (see salinity map in fig. C1 ). [...] We consider phytoplankton cells that are stranded out of the water by the receding tide, and lie dry for more than 7 consecutive days to be dead and remove them. [...] Phytoplankton cells will also die if they are light-limited for 14 days. This value is based on measurements presented in (Walter et al., 2017) [...] They are considered light-limited below a depth of 1m based on SPM data presented in (Stanev et al., 2019). The initial batch of phytoplankton cells starts their life with a full light budget of 14 days, and each minute below 1m reduces this budget by one minute, while the opposite applies if they are above 1m. When a cell splits both inherit the same remaining light budget.

The section on particle counts now reads:

In both sets of experiments, we release 10,000 individuals representing the studied phytoplankton population at the beginning of the year. This results in over 1 billion individual particles simulated for each case with approximately 1 million particles active simultaneously counted over all cases over 500,000 time steps. This corresponds approximately to a one to one ratio of simulated phytoplankton cells to mesh nodes in the hydrodynamic model at each time step

- **Provide a sensitivity analysis of the many parameters of the model, in particular those concerning phytoplankton demography like mortality, reproduction rates, etc...**

We have added a sensitivity analysis for the mortality conditions of phytoplankton dry-out and light limitation to the sensitivity analysis already performed for growth rates and vertical velocities in the appendix. Varying these parameters changes the break-even point of growth and loss as

expected but no regime shift occurs and the observed trends remain the same.